# "The algorithm will screw you": Blame, social actors and the 2020 A Level results algorithm on Twitter

Dan Heaton[1]*, Elena Nichele[1,2], Jeremie Clos[1], Joel E. Fischer[1]

1 School of Computer Science, University of Nottingham, Nottingham, Nottinghamshire, United Kingdom,
2 Lincoln International Business School, University of Lincoln, Lincoln, Lincolnshire, United Kingdom

* daniel.heaton@nottingham.ac.uk

## Abstract

In August 2020, the UK government and regulation body Ofqual replaced school examinations with automatically computed A Level grades in England and Wales. This algorithm factored in school attainment in each subject over the previous three years. Government officials initially stated that the algorithm was used to combat grade inflation. After public outcry, teacher assessment grades used instead. Views concerning who was to blame for this scandal were expressed on the social media website Twitter. While previous work used NLP-based opinion mining computational linguistic tools to analyse this discourse, shortcomings included accuracy issues, difficulties in interpretation and limited conclusions on who authors blamed. Thus, we chose to complement this research by analysing 18,239 tweets relating to the A Level algorithm using Corpus Linguistics (CL) and Critical Discourse Analysis (CDA), underpinned by social actor representation. We examined how blame was attributed to different entities who were presented as social actors or having social agency. Through analysing transitivity in this discourse, we found the algorithm itself, the UK government and Ofqual were all implicated as potentially responsible as social actors through active agency, agency metaphor possession and instances of passive constructions. According to our results, students were found to have limited blame through the same analysis. We discuss how this builds upon existing research where the algorithm is implicated and how such a wide range of constructions obscure blame. Methodologically, we demonstrated that CL and CDA complement existing NLP-based computational linguistic tools in researching the 2020 A Level algorithm; however, there is further scope for how these approaches can be used in an iterative manner.

## 1 Introduction

Blame and agency in relation to automated decision-making is an emerging topic in academia [1]. Although currently under-explored, studying this has shown to be important when forming interventions for when decision-making algorithms do not do their intended job [2]. A recent example of this is the case of the 2020 A Level algorithm in England and Wales, where examinations during the Covid-19 pandemic were replaced by automatically calculated grades.

**Data Availability Statement:** All relevant data are within the paper and its Supporting information files.

**Funding:** All authors are supported by the UKRI Trustworthy Autonomous Systems Hub (UKRI

Grant No. EP/V00784X/1) (https://gow.epsrc.ukri.org/NGBOViewGrant.aspx?GrantRef=EP/V00784X/1). Dan Heaton is supported by the Horizon Centre for Doctoral Training at the University of Nottingham (UKRI Grant No. EP/S023305/1) (https://gow.epsrc.ukri.org/NGBOViewGrant.aspx?GrantRef=EP/S023305/1). The funders had no role in study design, data collection and analysis, decision to publish, or preparation of the manuscript.

**Competing interests:** The authors have declared that no competing interests exist.

Although initially defended, the algorithm-decided grades were abolished and teacher assessment grades were used instead due to an outpouring of public dismay.

Although work has been done on collecting public perspectives about the A Level algorithm, there is a research gap regarding public views expressed on Twitter, which could be a valuable source of data as it hosts a plethora of views relating to current affairs [3, 4]. Therefore, addressing this research gap could provide a fuller and more detailed picture of the wider public's response to the event. To date, one contribution by Heaton et al. [5] has examined Twitter discourses relating to decision-making algorithms—including the 2020 A Level algorithm through the use of computational linguistic approaches, including sentiment analysis, due to their popular use in analysing trending topic discussions on Twitter [6–8]. Their analysis found sentiment fluctuated throughout the discourse, though was predominantly negative. In particular, fear and anger were the most prominent emotions, whilst discussions around the government, teachers and statistics were taking place.

However, due to the limitations of this approach—such as interpreting results and inconsistencies when comparing sentiment scores to human review [9, 10]—we build on this through the use of Corpus Linguistics (CL) and Critical Discourse Analysis (CDA). This qualitative analysis is underpinned by Social Actor Representation (SAR), a branch of Social Action Theory (SAT), where grammatical and transitivity structures play a crucial role in the representation of social actors [11]. Transitivity analysis—the examination of active and passive agents in texts—may uncover who is acting as the agent over whom and whether passive verbal constructions delete or mask social actors. There are various SAR techniques that indicate whether an agent in a text is a social actor, including *exclusion*, *backgrounding*, *individualism*, *assimilation*, *personalisation* and *impersonalisation*, which will all be explored. Thus, using SAR is helpful when examining blame and responsibility in discourse.

Our contribution is based on the belief that applying CL and CDA to Twitter discourses can mitigate some of the potential shortcomings of NLP-based computational linguistic tools [12]. This is due to the high emphasis on context and how language is used, underpinned by SAR. In fact, studies into Twitter discourses using these methods have yielded insightful and meaningful results on women driving in Saudi Arabia [13], refugees [14] and the dislike for hyperfeminized items being marketed to women and girls [15]. These examples showcase how this approach can be used in the wider context of social media research, which will be examined in this contribution.

This paper intends to add to the original findings from Heaton et al., which were affected by the shortcomings of the approaches discussed above. Opportunities offered by CL and CDA will be explored in this work, which shares the same dataset as Heaton et al., ultimately digging deeper into the discourse and finding out who Twitter users blame for the disruption to A Levels. Ultimately, using this combined approach will add to the the current discourse regarding which entities have been blamed for the algorithm's failure, particularly illuminating ideas about how social media users reacted to the scandal.

Summarising, this paper will use CL and CDA to examine how blame is implied in relation to automated decision-making, through agency and transitivity, in Twitter discourses regarding the A Level algorithm. From a practical perspective, the entities will be identified through the aid of SAR. From a theoretical perspective, complementing NLP-based computational linguistics with CL and CDA will illustrate a hybrid language analysis approach.

## 1.1 Context of the 2020 A Level algorithm

On August 13th 2020, Ofqual (The Office of Qualifications and Examinations Regulation), the UK examinations regulations body, used a decision-making algorithm to replace the standard

A Level qualifications, which had been cancelled that year due to the Covid-19 pandemic. The algorithm—defined here as the processing of data to produce a score through classification and filtering [16]—used prior centre attainment and teacher assessments to generate a grade for each qualification [17]. In comparison to the predicted outcomes submitted by their teachers, 35.6 per cent of students had qualification results lowered by one grade, 3.3 per cent by two grades, and 0.2 per by three grades [18]. The conditions that their university offers or employment opportunities were required were unmet. Therefore, their career plans were irreparably compromised.

This became a highly contested issue to schools, regulators and the wider public [19]. The key aspect criticised was that prior assessment data and teacher-assessed grades had been submitted but not used in their sole form [20]. Instead, they were combined with previous assessment data. That rendered the calculation unfair to students and educators from high deprivation communities especially.

The UK government defended the use of the algorithm initially, as it helped combat grade inflation. However, due to public outcry, it retracted the algorithm-generated grades on August 17th 2020. Instead, all qualifications were awarded the teacher-submitted grades [21]. The Education Secretary of State at the time, Gavin Williamson, appeared to place blame on Ofqual and emphasised he was not aware of the scale of the problem [22]. The public reaction also saw the resignations of Sally Collier, CEO and Chief Regulator of Ofqual, and Jonathan Slater, the most senior civil servant in the Department for Education. Therefore, the social impact of the choice went well beyond the class of 2020.

Ofqual reported there was no grading bias [23]. However, it was found that the algorithm favoured students from more economically privileged backgrounds while other suffered more [24]. This was due to each school's historic results being a significant factor in the algorithm's grade calculation. This led to the algorithm being labelled as 'mutant' by UK Prime Minister Boris Johnson [25]. Ofqual officials were quick to blame 'overly generous teachers', but not the algorithm itself [19].

Several studies examined the impact of the algorithm. Bhopal and Myers surveyed 583 students and interviewed a further 53 students who were eligible to take A Level examinations, between April and August 2020 [26]. Their aims were to to examine the impact (mental and academic) of predicted grades on A Level students, explore support systems in place for such students, and analyse differences by race, class, gender and school type. Through quantitative and qualitative analysis, it was found that students had identified the significance of unfairness within their individual experience. Students from all types of school and background felt the deployment of the algorithm placed little or no value on individual students' experiences. Consequently, many students received results they perceived to be unfair (21% of those surveyed said they were happy with their results), which was in contrast to the official investigation report that concluded that there was no grading bias [23].

Additionally, Kolkman noted that the incident shone a light on algorithmic bias [27]. However, he also noted that greater knowledge of algorithmic-driven decisions requires better understanding of the functionality. More specifically, the author foregrounded the importance of critical reflection within the process of algorithm design and noted that, without intervention, there will be further unrest and distrust in algorithms that impact daily lives. Hecht further examined the social impact of using the algorithm [28]. They stated that public awareness, scrutiny, and transparency are critical first steps to eliminate perceived bias from the algorithm but far from a guarantee. Therefore, these are important factors to consider when examining views expressed about the algorithm. Ultimately, the current literature demonstrates that different entities have been blamed for the algorithm's failure, yet

limited research into how social media users reacted to the scandal, thus providing motivation for our research.

As indicated previously, only one study has taken social media responses into account when considering the public reaction to the algorithm [5]. However, the NLP-based approach, as well as issues with accuracy and interpretation, meant that the contribution did not explore social actors and who was therefore blamed. As a result, we propose using CL and CDA, underpinned by SAR, to examine who users portrayed as social actors and who was blamed. The following section will look in more detail about the shortcomings of NLP-based computational linguistic tools. It will also examine how CL and CDA, underpinned by SAR, can achieve more detailed insights into who users presented as social actors, and therefore blamed, through the exploration of grammatical agency.

## 2 Related work

To demonstrate the need to combine the aforementioned analytical approaches, an overview of limitations affecting sentiment analysis, among other approaches, will follow. This is set up in the context of the previous Heaton et al. study [5]. Additionally, an outline of CL and CDA —the chosen approaches—will be used to review existing contributions, which used similar methods to investigate Twitter discourses. Using CL and CDA, underpinned by SAR, it will be possible to ultimately contribute to filling the gap previously identified in the literature by identifying social actors in the Twitter discourse, providing an indication of who social media users blamed for the assignment of A Level grades in 2020.

### 2.1 NLP-Based computational linguistics to examine social media

Popular NLP-based computational linguistic tools can support the identification of the viewpoints expressed in large social media datasets. Sentiment analysis can offer such insights through predictive algorithms, which work on a binary polarity scale [29, 30]. For example, Park et al. used VADER, a sentiment analysis tool, to investigate fashion trends on Instagram and proved that the strong social media presence of a model was more effective than a contract with a top agency [31]. Similarly, Sivalakshmi et al. explored the sentiment towards the Covid-19 vaccine using TextBlob, another sentiment tool, and concluded that the discourse was neutral-to-negative in polarity [32].

As previously mentioned, Heaton et al. used sentiment analysis and other NLP-based computational linguistic tools to examine the views expressed on Twitter regarding the Ofqual algorithm, [5], critically evaluating sentiment analysis, topic modelling and emotion detection tools for textual analysis purposes.

Their findings showed that, from the TextBlob sentiment analysis, Fig 1 indicates that overall sentiment ranged from 0.088 to -0.052 and that overall sentiment was neutral. However, from the VADER sentiment analysis, Fig 1 shows that overall sentiment ranged from 0.03 to -0.5, indicating that overall sentiment was negative.

Additionally, they also found that the sentiment analysis in Fig 1 showed negative change in sentiment on August 14th, the day after the results were shared with students. On 17th August, when the government reversed the decision, there was a rise in positive sentiment. Although, on 26th August, when then-UK Prime Minister Boris Johnson told students that their results had been affected by a 'mutant algorithm'—mentioned in the introduction –, a sharp negative change occurred, potentially caused by the negative word 'mutant' and associated negative terms used in response to this. On September 3rd, when Ofqual Chair, Roger Taylor, apologised to students when appearing at the Educational Select Committee at the House of Commons, another positive rise could be seen.

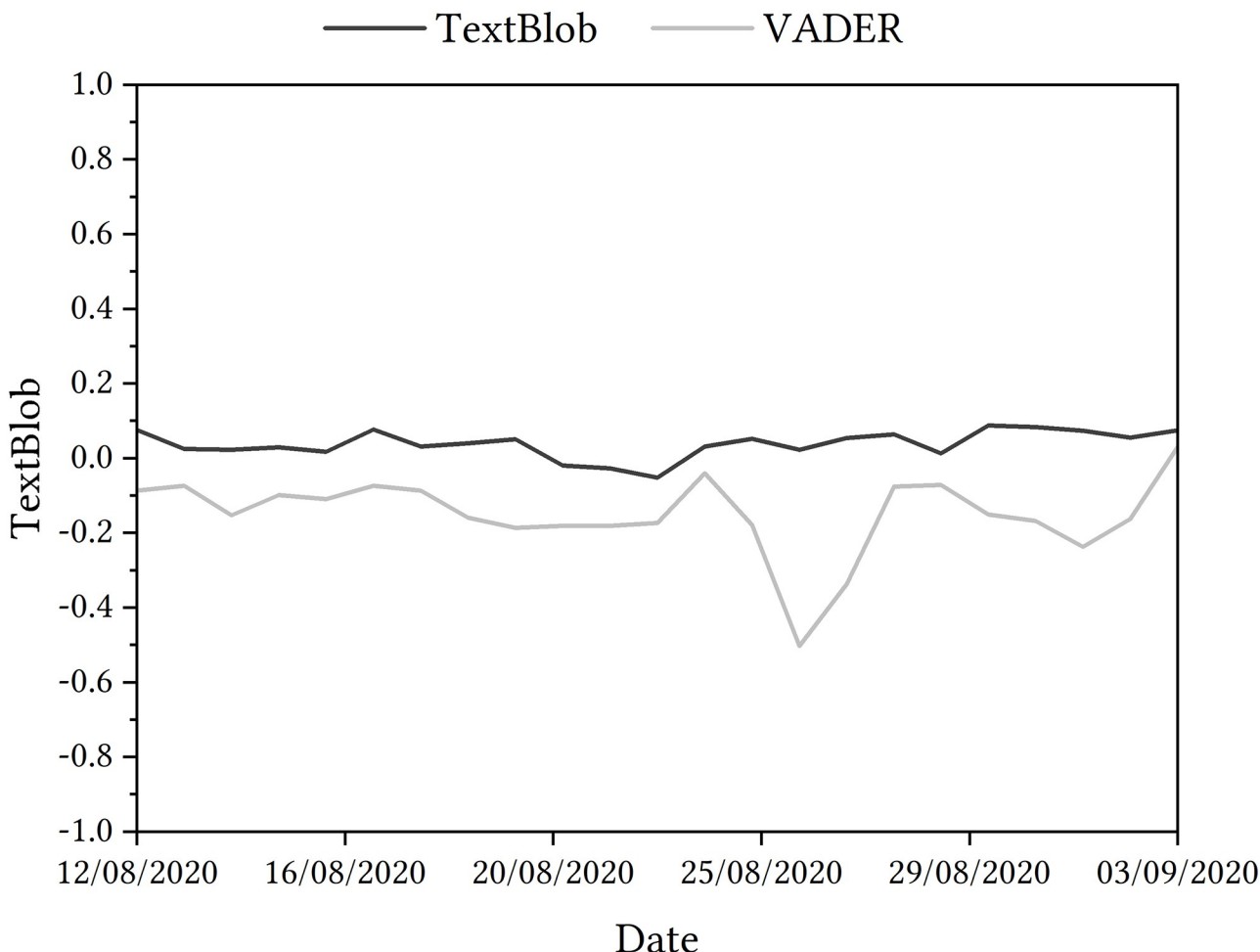

**Fig 1. Sentiment analysis of tweets relating the A Level algorithm in 2020 by Heaton et al. [5], licensed under CC BY 4.0.**

Additionally, other findings reported were that the most featured word of the most prominent topic was 'government', foregrounding their role in using and then withdraw the algorithm. 'Trust' was the emotion detected most frequently, but the direction of trust was not clear. Although this was a good starting point to capture general trends, these results struggle to explain why changes occurred or who tweet authors blamed.

As previously mentioned, sentiment analysis struggles to detect nuanced opinions. Notably, Heaton et al. [5] found difficulties in aiding their interpretation and potential applications in their study. This is echoed in similar studies [9, 10, 33]. Therefore, these findings might benefit from more rigorous qualitative analysis to unearth nuance and detail, especially when it comes to blame.

Generally, as well as this, experts have sought to combine computational linguistic tools with other methods to mitigate these shortcomings. These combinations have ranged from manual inspections [34] to comparing human and algorithm classification [35, 36]. Human analysis provided the most accurate results, illustrating the need to combine these computational linguistic tools with other approaches for validation. However, these tools still do not clearly identify grammatically and social actors or who is blamed. Therefore, we aim to mitigate this challenge by incorporating CL and CDA.

## 2.2 Using Corpus Linguistics to examine social media

One suitable approach to provide insight is Corpus Linguistics (CL). A corpus is defined as a body of written text or transcribed speech, which can be linguistically or descriptively analysed [37]. CL takes this idea of further investigating the corpus through a multitude of different analytical tasks. This is the study of language data on a large scale [38]. CL allows for the comparison of multiple corpora (more than one dataset) to identify trends and patterns in a texts, which is particularly helpful when comparing data from different time periods, such as in this study.

As data is tagged according to the part-of-speech (noun, verb, adjective, etc.), analysis can begin. One of these analytical methods is collocation. Collocation is defined as the co-occurrence of two or more words within a defined word span [39]. When using frequency as the sole measure, Baker states that it might not be possible to verify whether a co-occurrence is a true reflection of a semantic relationship or whether chance played a part [40]. Instead, statistical significance measures, such as LogDice (or Log Likelihood), become a useful indicator of lexical and grammatical associations between textual elements, as well as themes [41]. In this sense, concordances help identify collocations as they can show how adjacent or in close vicinity the related words are together. Therefore, concordance lines can display the context surrounding a word of interest [42].

There are advantages to using CL to analyse social media datasets. According to Jaworska, CL offers an ease in how large amounts of data can be automatically scanned to uncover patterns in frequency and keywords [39]. This is echoed by Tognini-Bonelli, who states that CL allows access to real-world, authentic texts and a high processing speed [43]. Given its efficiency and capacity to process large datasets, CL facilitates diachronic comparisons across corpora through lexical usage [44]. Because of its capacity to point out language patterns in large datasets, CL has been frequently deployed to carry out analyses on social media.

Jaworska also categorises media research involving CL into two strands: the first focuses on structural, pragmatic and rhetorical features of text, and the second on how language shapes representation [39]. Similarly, Nugraha et al. concentrated on both whilst investigating a Twitter corpus about the 2020 Charlie Hebdo shootings, the terrorist attacks to the headquarters of the French satirical magazine [45]. While '#JeSuisCharlie' was used to most frequently express sympathy, '#CharlieHebdo' featured in messages dealing with a wider variety of topics and emotions. Through using keyword and concordance analysis, and building on the previous CL findings of Kopf and Nichele [46], they found that there were 13 categories of keyword—such as place, the weapon, and the attacker. These categories are connected to each other: for example, many tweets linked the attacker to Islam, his religion, and discussed Pakistan and Islamic culture generally, framed by this incident. These studies all constitute examples of using CL to analyse Twitter discourses of social interest or having an impact on society.

Despite its key advantages, CL can pose analytical challenges with social media data. For instance, Baker found that CL, used in isolation, provides a focus on collocation and word frequencies, which is descriptive in functionality, and thus focus is drawn away from interpretation or critique [47]. Rose also criticises the restricted explainability of CL-derived results, despite the large evidence these could provide [48]. In this sense, the author calls for an integration of CL with other qualitative approaches to ensure more meaningful insights. These recommendations appear supported by Sulalah, who investigated the semantic prosody of 'increase' in Covid-19 discourses [49]. Additionally, Liimatta states that CL analysis can be problematic when dealing with short texts because of its normalised counts—usually calculated on a base of either 1,000 or 10,000 [50]. The calculations could generate unreliable values when applied to very short texts—such as tweets—due to the excessively small lexical samples

these allow to consider. As a result, very short texts, which are especially common on certain social media platforms, should be interpreted carefully when compared.

## 2.3 Using critical discourse analysis to examine social media

Considering the challenges posed by CL, discussed in the previous section, we chose Discourse Analysis (DA) as a complementary approach. Whilst CL analysis tools struggle to pinpoint different perspectives and meaning shades, DA examines texts for nuance and pragmatic opinion (here meaning an examination of implied meanings of language). Therefore, these approaches were deemed especially effective together to explore blame in the A Level algorithm Twitter discourse.

Discourse surpasses the sentence boundaries [51] and comprises language stretches that are interlinked and create meaning, thus they carry an inscribed sociolinguistic value [52]. In this sense, questioning the social significance of language can uncover how it influences –- and is influenced by—the world around us [53]. Therefore, DA is an interpretative qualitative approach to text analysis that draws upon related theoretical frameworks.

In fact, there are several foci that can be adopted when approaching DA and Hodges et al. label them as descriptive, empirical and critical [54]. While descriptive addresses solely how language and grammar work together to create meaning in isolation, empirical and critical variations account for context and even include it as part of the data collected from discourses. Empirical analysis has been used successfully in studies where there is still a microanalytical focus on language. However, critical analysis places even greater emphasis on contextual information through macroanalysis, which focuses on the power and perspectives of individuals and institutions. As this is relevant to our aim, we will apply Critical Discourse Analysis (CDA) in this paper.

CDA can be used as a tool to better understand meanings implied by the context of a text or series of texts [55]. Fairclough identifies three CDA layers: micro, meso and macro [56]. Micro analysis examines syntax (sentence construction), metaphorical meanings and rhetoric. Meso analysis looks at the interpretation of the relationship between discursive processes and the text. Macro analysis examines the explanation of the relationship between the discourse and the socio-cultural reality that is external to the text.

Another significant contribution with regards to contextual meanings of the discourse is put forward by Van Dijk, who offers a socio-cognitive perspective [57]. Accordingly, discourse can be viewed as socially shared representations of societal arrangements, as well as interpreting, thinking, arguing, inferencing and learning. Although different, the two contributions are similar in regards to transitivity [58]. For example, an examination of transitivity patterns may uncover who is acting as the agent—thus, performing the action—over whom and whether passive verbal constructions exclude and background social actors. Therefore, this shows existing studies employing CDA show that a specific focus on agency is possible to unveil blame and responsibility.

More specifically, Leslie defines an agent as an entity with an internal source of energy through which it exerts force supposedly to carry out the action referred in the text [59]. Expanding on this, Richardson et al. state that that agency in linguistics is often explored by examining how it is emphasized, manipulated, or concealed [60]. As such, transitivity analysis —the examination of agency in text—looks at the use of active and passive voice or the nominalization, where verbs are word class converted to nouns. Here, choices reveal the attitude and ideology of the language user.

Additionally, research shows that passive constructions tend to remove agency from the subject. Especially when the subject is absent from the clause, there are shifts in blame [61].

Alternatively, agency can be implied through lexical choices. For instance, Morris et al. suggest that that "acceding trajectory evokes impression of high animacy, which would be caused by enduring internal property, i.e. the volitional action" (e.g., "the NASDAQ fought its way upward") [62]. On the other hand, "the descending trajectory suggests inanimacy, as a result of lack of external forces." (e.g., "stocks drifted higher").

Metaphors have also been used to personify inanimate entities and increase the dramatic effect and intensity of a statement [63]. Additionally, vocabulary can be examined to unearth how words are used to show ideology, including the use of euphemisms and metaphors. It is also important to factor in how implicit information can be inferred and deduced through the examination of these aspects of language. Given its relevance, this work will use transitivity and agency as a focus of our analysis.

Similar studies have used CDA to examine Twitter data, whilst addressing other social aspects such as gender and origins. Among them, Aljarallah et al. investigated perspectives on women driving in Saudi Arabia, finding specific hashtags that were supported or opposed to women driving [13]. Their results showed, among others, that tweets with the hashtag #Womencardriving presented significant support towards the movement. However, opposing reactions emerged from the hashtags #Iwilldrivemycar and #Iwillentermykitchen. In another study by Sveinson et al., representations of gender and stereotyping have also been explored, including overwhelming dislike for hyperfeminized items marketed to women and girls through detailed linguistic analysis [15]. This study demonstrated that fan clothing serves as more than just a reflection of consumer preferences, as it can also embody the cultural identity of an organisation. Also, Kreis investigated the hashtag #refugeesnotwelcome, unearthing that users deployed a rhetoric of inclusion and exclusion to depict refugees as unwanted, criminal outsiders [14]. Her findings showed that this discourse reflected a prevailing political climate in Europe, where nationalist-conservative and xenophobic right-wing groups were gaining influence and promoting a discourse that is prominent on social media. Overall, these studies demonstrate the benefits of using CDA on Twitter discourses specifically, highlighting the depth of understanding that it can uncover.

Notably, CDA brings several advantages as it can reveal unacknowledged aspects of human behaviour and support new or alternative positions on social subjects [12, 64]. In this sense, CDA is naturally interdisciplinary [65] and requires an adductive approach, where a symbiotic relationship between theory and empirical data is necessary [12]. As CDA examines the intricate relationships between text, social opinion, power, society and culture, it provides a lens to better understand urgent social implications [55]. Additionally, the incorporation of an epistemological aspect into CDA means that, while the researcher brings their own beliefs and perspective, reflection upon findings has its place within the approach. Bucholtz claims this to be reflexivity with a heightened self-consciousness [66]. Therefore, CDA is an appropriate choice to explore social action, blame and agency, as in this study.

As with any methodological approach, CDA has shortcomings, too. Firstly, it requires considerable effort and time required to perform CDA on a large dataset [67]. Additionally, the subjective nature of CDA, approaching data with a personal perspective and lens, may limit its validity and decrease the objectivity and applications of the findings [68]. Both shortcomings provide a case for combining computational linguistic analysis with CDA. Also, Morgan notes CDA is not fixed and is always open to interpretation and negotiation [64]. The lack of objective measures available to analysts may result in inaccurate or misrepresentative findings. This complements the view of Olson that it is not a 'hard science' and more of an insight through examination and discussion [69].

These shortcomings provide a rationale for using CL with CDA to increase processing efficiency. This also aids the mitigation of the potential subjectivity of CDA: using a semi-

automated approach first means comparisons can be organised according to the research focus. Although combining CL and CDA does not grant ultimate objectivity, it is less prone to exclusive subjective analysis.

## 2.4 Research gap

As previously discussed, combining all three of the approaches—sentiment analysis, CL and CDA—is uncommon. Nevertheless, existing contributions have demonstrated the individual efficacy of each to analyse features and characteristics of specific social media discourses. Thus, this work sets out to test whether their combination could provide a more complete account into the agency of potential social actors within the A Level grade calculation discourse. We aim to fill this current research gap by underpinning our analysis with Social Actor Representation (SAR), drawn from Social Action Theory (SAT).

SAT states that "people create society, institutions and structures" [70]. According to Engestrom, examining social actions can provide an explanation for human behaviour and societal change [71]. In other words, SAR is a branch of SAT which examines how grammatical structures convey social agency. For example, active or passive constructions and transitivity structures can be employed to communicate who social actors are in discourse [11].

Moreover, references to grammatical agents do not necessarily need to be present in discourses altogether. This choice is called *excluding*, and, in *backgrounding*, clues can be left in. Other strategies include *individualism*, which implies referring to actors as individuals or *assimilation* by referring to actors as groups. Also, actors can be *personalised* through word choices pertaining to the semantic nature of being 'human' or *impersonalised*. All of these representation structures play a role in indicating the social and power dynamics within discourse, as shown in other Twitter case studies that used CL and CDA [72–74]. For example, McGlashan explored the language patterns of followers of the Football Lads Alliance, revealing correlations between follower profile descriptions and their tweets, indicating a construction of identity tied to radical right-wing and populist discourse regarding Islam where Islam is attributed agency [72]. Moreover, Fadanelli et al. found that social actors in former Brazillian president Jair Bolsonaro's pre-campaign and government tweets served to publicise the president's enemies, promote polarization, and align with his ideology, ultimately impacting his popularity among supporters both positively and negatively [73]. Finally, Bernard studies the construction of social actors in the reports of two South African mining companies, revealing how linguistic representations of higher- and lower-wage employees contribute to power dynamics and social inequality in the industry, which emphasised the agency of these companies in shaping relationships and maintaining dominance. These studies indicate the potential value of combining SAR with CL and CDA.

Finally, it is important to note that the application of SAR illuminates insights through a novel perspective that would not have been possible using popular NLP-based computational linguistic tools alone. Therefore, using SAR with CL and CDA will allow further unpacking of the social implications discovered in this Twitter discourse.

## 3 Method

As previously mentioned, using Twitter has allowed the collection of a large, readily available dataset. Twitter data can be processed before analysis [75], lending itself well to exploratory analyses [76].

For convenience, data was collected using the Twitter for Academic Purposes Application Programming Interface (API) and Tweepy [77]. We ensured that the collection and analysis method complied with the terms and conditions for the source of the data and the API. The

data were sourced from the United Kingdom and only tweets in English were selected, meaning the analysis investigated views expressed in English only. Since retweets indicated agreement or support, duplicate tweets were expected, although eliminated from the corpora not to bias counts.

The 18,239 tweets composing the dataset were published from 12th August 2020, the day before A Level results were released to students, until 3rd September 2020, after Ofqual's chair appeared at the Education Select Committee. Tweets containing 'Ofqual algorithm', 'ofqualalgorithm', 'A level algorithm', 'alevelalgorithm', 'a levels algorithm', 'a-level algorithm' or 'a-levels algorithm' were gathered. These search terms were chosen on the basis of their relevance to the algorithm, rather than the A Level results in general. The tweet IDs, and other associated information, can be found in S1 Dataset.

The next step concerned CL. Using the CL software The Sketch Engine [78], a keyword analysis was conducted to investigate frequently featuring social actors. The reference corpus used was the English Web 2020 (enTenTen20) [79], which comprises of 36 billion words of internet texts. Since it contains texts from social media, this was believed to be a suitable reference corpus for this study.

Firstly, comparing our corpus to the reference corpus is used to generate a keyness score, which was calculated by comparing the frequency of the words in the target corpus to the frequency of the words in the reference corpus. Secondly, concordance lines featuring potential social actors were examined to prompt the collocation analysis. This included using LogDice as a statistical measure of collocational strength. Thirdly, CDA was used to examine agency and blame as expressed in the concordance lines, where the selected keywords appeared in context.

Additionally, the focus was placed on transitivity, through the examination of social actors in sentence structures, vocabulary choice and the use of metaphor and possession. Specifically, principles of Leeuwen's SAR were employed to provide insight into these social representations. Therefore, we looked at items of interest that could be related to blame, agency and social action through the collocation analysis of their concordance lines.

Despite the advantages just discussed, using tweets as a reflection of specific social media discourses carries risks [80]. Firstly, complex ethical considerations have to be made when scraping data for analysis from Twitter. For instance, a prominent ethical issue is the fact that although tweets are public by default, Twitter 'data' is not actively provided by users for research purposes, yet gaining explicit consent to use tweets from their authors is practically unfeasible [81]. Therefore, we decided not to attribute to any specific excerpts of tweets, mentioned in the results section, trusting they could hardly be attribute to specific users, as approved by the university department's ethics committee. As an extra precaution, data was pseudonymised during the extraction process, with a unique number generated by the Twitter Academic API referring to each tweet. Hence, this is why only tweet IDs are available in S1 Dataset, rather than the tweets in their entirety.

## 4 Results

This section first comprises of the CL keyword analysis, which led us to identify potential social actors for investigation. Based on this first list, four potential social actors (the algorithm, Ofqual, the government and students) were investigated through the examination of collocational strength and CDA.

### 4.1 Keyword analysis of potential social actors

Table 1 shows the top ten words with the highest keyness score when compared to EnTenTen2020. From this analysis, the main findings were that four potential entities were identified:

**Table 1. The top ten words with the highest keyness score.**

| Item | Relative frequency (per million) | | Score |
|---|---|---|---|
| | Focus corpus | Reference corpus | |
| algorithm | 28,339.45 | 0.51 | 29.3 |
| a-level | 9,881.38 | 1.27 | 10.9 |
| ofqual | 8,598.08 | 0.08 | 9.6 |
| results | 6,261.83 | 14.79 | 7.2 |
| grades | 5,717.25 | 7.94 | 6.7 |
| students | 4,730.1 | 94.14 | 5.2 |
| a-levels | 4,175.65 | 1.61 | 5.2 |
| by | 7,584.61 | 471.41 | 4.9 |
| exam | 2,826.54 | 30.25 | 3.7 |
| government | 2,518.88 | 45.21 | 3.4 |

the algorithm itself, Ofqual, students and the government, as they all appeared as keywords. These were identified as they were all nouns that had the potential to be presented actively in a grammatical construction, thus could be a social actor. The following sections detail how blame is placed or not placed on the entity of concern through the main events of the discourse.

## 4.2 The algorithm

Collocational strength of the top ten words associated with *algorithm* is shown in Table 2 (after stopword associations were removed). The trajectory of the collocations over time can be seen in Figs 2 and 3. Both *a level* and *ofqual* appeared as adjectival modifiers to *algorithm*. *Flaws* collocates strongly with *algorithm* at the start of the discourse, pertaining to one particular tweet that had been retweeted many times about a father (hence the strong collocation with this word too) that points out 'algorithm flaws'. This returned towards the end of the discourse, where there were many tweets discussing how Education Secretary Gavin Williamson 'knew of the flaws of the algorithm'. Words with high collocational strength that are in the semantic field of education, such as *results*, *grades* and *exam* were also present, but could not tell us much about how the algorithm was presented. Therefore, from this analysis alone, it is not clear whether the algorithm itself had grammatical agency grammatical or perceived social agency from tweet authors.

**Table 2. Collocational strength of *algorithm*.**

| Collocate | Freq | Coll. freq. | logDice |
|---|---|---|---|
| a-level | 3405 | 6006 | 12.2297 |
| ofqual | 2393 | 5226 | 11.7701 |
| results | 1324 | 3806 | 11.0105 |
| flaws | 1090 | 1149 | 10.9247 |
| a-levels | 1110 | 2538 | 10.8458 |
| foresaw | 997 | 1015 | 10.8066 |
| father | 991 | 1025 | 10.7971 |
| exam | 1004 | 1718 | 10.7622 |
| grades | 1053 | 3475 | 10.703 |
| level | 903 | 1305 | 10.641 |

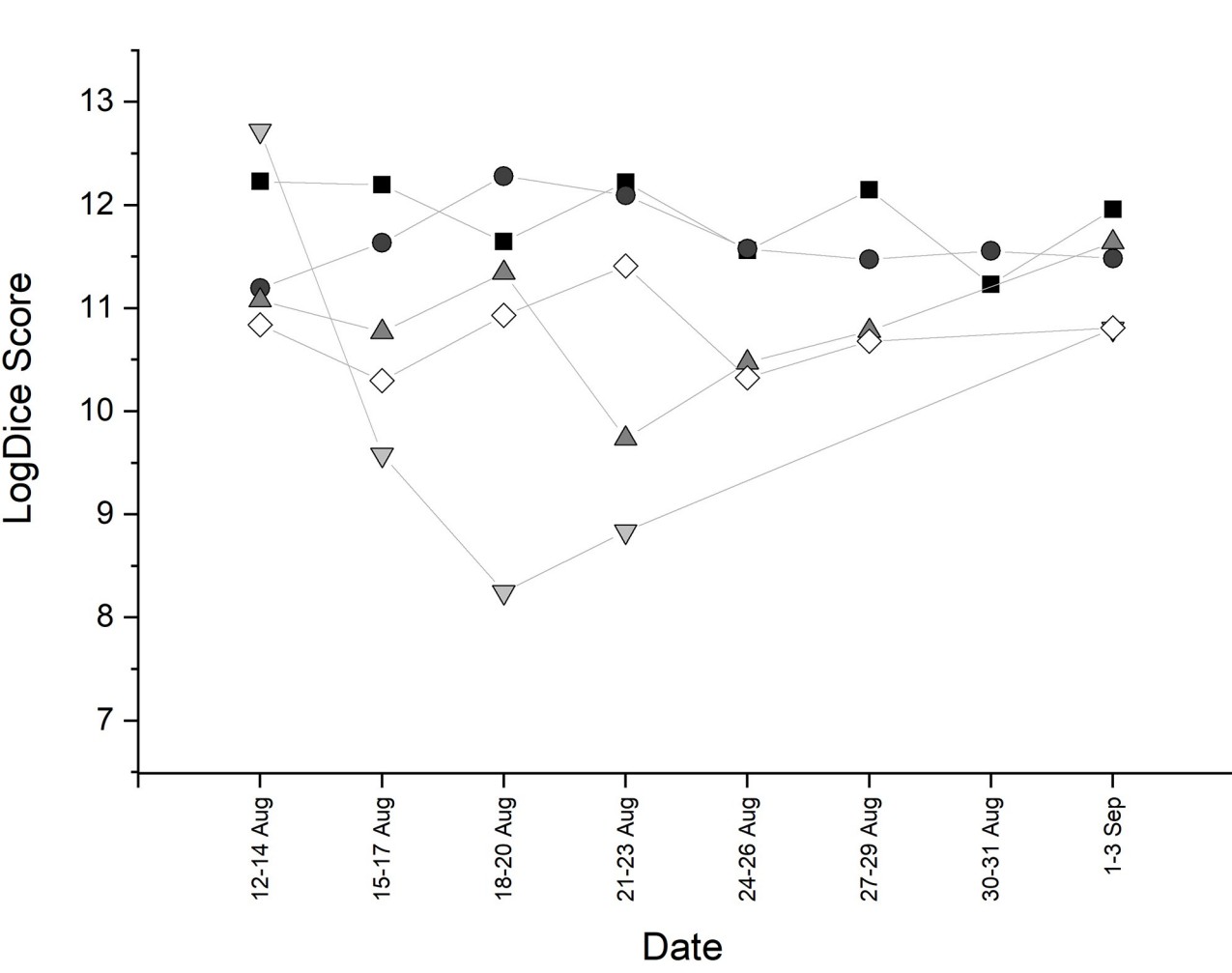

**Fig 2. Temporal trajectory of LogDice scores of collocates of *algorithm*—Part A.**

However, through the manual examination of other concordances, the algorithm itself is presented as having agency and potentially being blamed for the events that occurred. In this section, the key findings relate to the active presentation of the algorithm, its metaphorical agency and personalisation, and how this changes through the timeline as tweets show an undetermined responsibility for the actions.

On August 12th 2020, tweets show the algorithm performing a task as the social actor in grammatical constructions. Tweets that contain structures such as 'that algorithm is going to screw you' and 'this algorithm appears to be cementing that bias towards the wealthy' received a 235 total engagements (combined likes and retweets). The active syntactical structures implies that social agency is with the algorithm. On 13th August, the day results were released to students, there were also many tweets that gave the algorithm social agency, presented in a similar way, illustrated by the active statements that the algorithm 'caused today's chaos' (5795 engagements). Here, personalisation is seen. This is in addition to a tweet that contained 'the algorithm used by ofqual can't be applied to small cohorts' (5517 engagements), here fore-grounding the importance of the algorithm, despite a lack of agency, through this passive

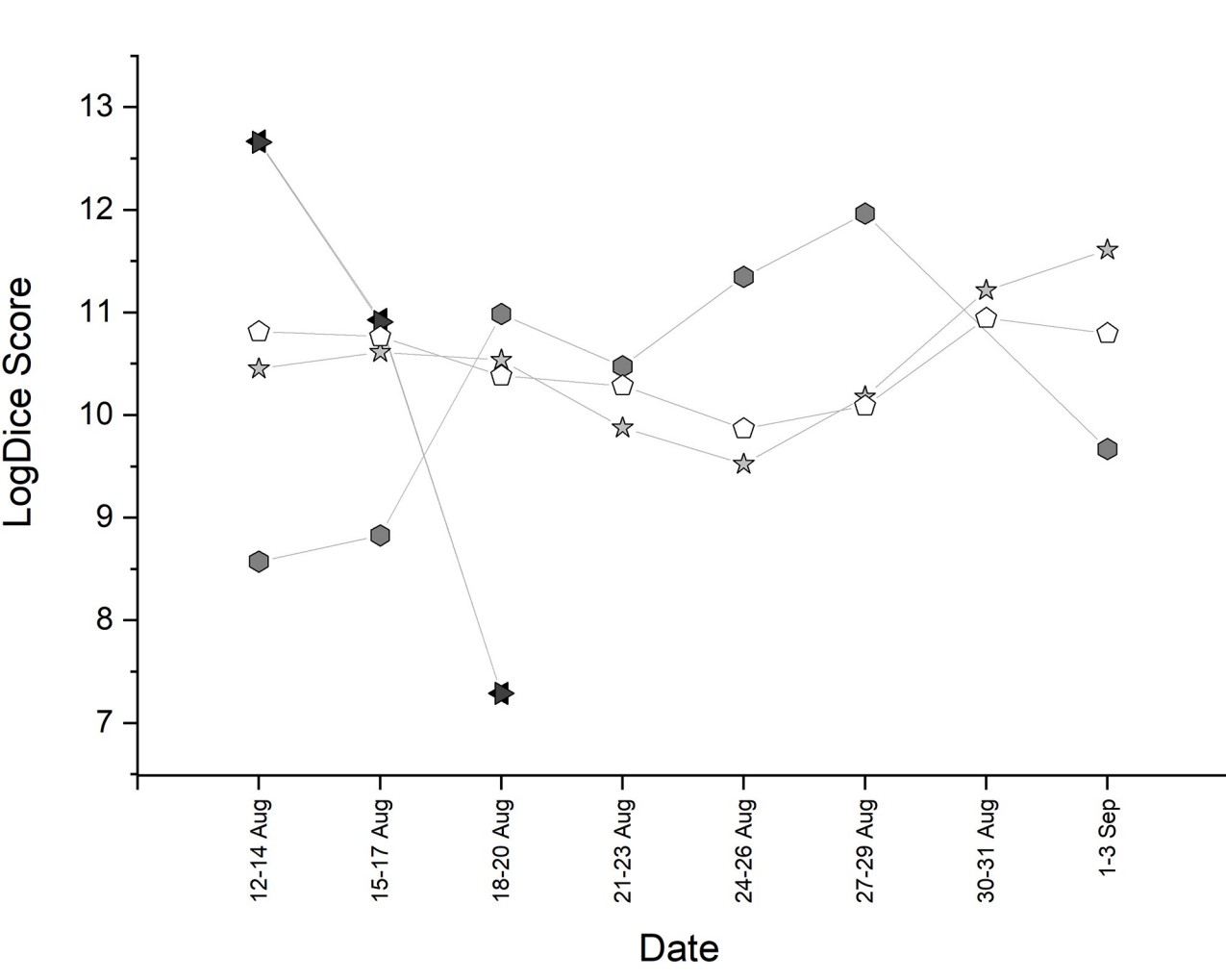

**Fig 3. Temporal trajectory of LogDice scores of collocates of *algorithm*—Part B.**

construction. This could be seen as the *backgrounding* of Ofqual and a foregrounding of the algorithm.

Prior to the government change, transitivity analysis shows more cases of the algorithm being presented in an unfavourable way. Regarding pathways to university, one tweet says that it is 'intolerable that an algorithm is denying this to others' (7774 engagements), a clear active grammatical construction that places agency with the algorithm. Another tweet states that 'this racist, discriminatory and downright evil algorithm is ruining lives' (2595 engagements)— overtly stating that the algorithm has the power to create significant impact on humans, thus being *personalised*. Additionally, a tweet on 16th August stated that '97% of gcse results fully decided by an algorithm' (1490 engagements). This implies that the algorithm has the capacity to make decisions on the outcome of the GCSE qualifications of students. Another well-engaged tweet on 16th August stated that the 'algorithm has given them Us and fails' (13256 engagements)—placing agency with the algorithm through *personalisation*.

This sentiment continued into the date of the reversed decision, 17th August 2020. One tweet with 2136 engagements included the clause 'your future should be based on your abilities

not an algorithm', continuing the notion that the algorithm has the potential to change lives. Another tweet with 7126 engagements said that 'private schools had done better with the ofqual algorithm'. Despite being part of a prepositional phrase in this context, the algorithm is still mentioned when the foregrounded part of the tweet is concerned with inequality of results. However, the algorithm is nominally labelled as 'the ofqual algorithm'—thus, despite the active presentation of the algorithm, it is owned by Ofqual, thus potentially blurring the boundaries of blame and accountability.

There are occasions when the algorithm is referred to as being 'used' by an unknown actor. This is first seen on the most engaged-with tweet on 12th August, the day before results were released to students, which stated 'the algorithm used to grade a-level results is incredibly sophisticated' (4513 engagements). The fact that a transitive verb 'used' is chosen here without a named active social actor creates the impression that authors believe the algorithm is not to blame for the results, but the anonymous 'user' is. There are further instances where this occurs, such as 'algorithm used for a-level grades' on August 17th (1695 engagements).

The algorithm is also presented passively, implying removed agency. One tweet with 1329 engagements states that people 'benefited from [the] algorithm' on 13th August. Additionally, the most engaged-with tweet on 15th August (10311 engagements) discussed the importance of rectifying the situation prior to the release of GCSE results the following week, stating that the qualifications would also be 'assigned \*solely\* by another ofqual algorithm'. While this presents Ofqual as the possessor of the algorithm and could imply blame, the algorithm itself is performing the task of 'assigning' despite being an inactive entity. This is in addition to a tweet on the same day that explains '1/4 state school students were downgraded by the algorithm versus 1/10 private school students' (2931 engagements). Here, again, while a passive construction is used, the algorithm is not the focus of the construction; instead, the focus is shifted to the inequality of the 'decisions' that the algorithm made. Thus, while blame is not attributed to the algorithm through syntactical structures here, the subject matter of the tweet places blame on it through the foregrounding of this comparison. This *backgrounding* limits the agency that the algorithm has as a social actor but still implies blame.

Passive constructions continue on 18th August, where a UK university tweeted about supporting students 'who have been disproportionately affected by the a-level algorithm' (298 engagements). Again, while this is a passive construction, agency may still be attributed to the algorithm as it has performed an action that affected a human. However, it must be noted that the construction of the sentence foregrounds the students in this case.

Further on in the discourse, on the 25th August, there are tweets that imply the algorithm is doing a 'job', an activity usually performed by a human. One author wrote 'Ofqual guidance doesn't require them to moderate—that was the job of the algorithm'. This personification and *personalisation* of the algorithm could place further blame and agency on it as a distinct social actor. This in addition to a user who details that the algorithm had 'failed [their] daughter', thus implying that the algorithm had agency to perform such an action.

To summarise, the algorithm is mostly seen in active constructions that indicate agency is with it as a social actor. The personalisation and agency metaphor strategies seen in tweets also add to the indication that people see the algorithm as a social actor too. There are, however, instances where the algorithm is portrayed in passive constructions, although blame could still be interpreted. In the final dates of the dataset explored, more tweets directed blame through agency at Ofqual and the UK government. There are some active constructions that involve the algorithm, but the majority are centered around the organisations or individuals. These social actors will now be explored in more detail.

## 4.3 Ofqual

This section explores Ofqual as a potential social actor, with a specific focus on active and passive agency, agency metaphor and individualism of a defined entity within Ofqual, Roger Taylor. Collocational strength of the top ten words associated with *Ofqual* is shown in Table 3. The trajectory of the collocations over time can be seen in Figs 4 and 5. Once again, lexicon associated with education was present. Collocations of interest included *ignored*. This was seen throughout the discourse, such as the 14th August ('ofqual ignored offers of expert help with its algorithm') and 20th August ('ofqual ignored exams warning a month ago'). The use of the word 'ignored' here could be seen as significant as it places Ofqual as the active social actor in the tweet. *Have* was also collocationally strong, often performing as an auxiliary verb where Ofqual is the social actor ('ofqual have created an algorithm which just doesn't work', 'ofqual have downgraded', 'ofqual who have ruined young lives' and 'ofqual have favoured the unadjusted small cohorts'). *Used* is seen in constructions that are active ('ofqual has used an unequal algorithm') and passive ('the algorithm used by ofqual') throughout the discourse. There was a great deal of engagement with a tweet that stated '"ofqual exam results algorithm was unlawful, says labour'. Although not an examination of agency, the use of the adjective *unlawful* might be an indicator of blame.

Through further concordance examination, users showed other ways in which they blamed Ofqual. Immediately, it is clear that the process of *assimilation* is present in tweets pertaining to Ofqual due it being a group. One of the most common situations that this occurred was by attributing ownership of the algorithm to Ofqual, as seen in tweets that contained the phrases 'its algorithm', found throughout the discourse.

Upon the revision of results, Ofqual was mentioned more in the discourse as an social actor. This is seen in tweets that involve the possession of the algorithm and some that talk about Ofqual as a separate social actor. In tweets that do discuss Ofqual as owners of the algorithm, such as 'experts question how their algorithm could so blatantly favour private schools', seen on 17th August with 4274 engagements, this possession is clear. However, the algorithm here still has some sort of agency as it is the social actor doing the 'favouring'. This blurs the lines between who the social actor is and, therefore, who is to blame. This implication of multiple entities that presents, with the algorithm as the social actor but Ofqual as the possessor, continues the following day. This is seen in a tweet with 270 engagements that states 'the government knew ofqual's algorithm would disadvantage the disadvantaged'. This may result in blurred blame.

As previously alluded to, there are tweets that foreground Ofqual as the social actor, rather than as the owners of the algorithm. For example, one tweet with 2029 engagements on 20th

**Table 3. Collocational strength of *Ofqual*.**

| Collocate | Freq | Coll. freq. | logDice |
|---|---|---|---|
| algorithm | 2396 | 17225 | 11.7719 |
| exam | 299 | 1718 | 10.4625 |
| results | 330 | 3806 | 10.2255 |
| exams | 227 | 1110 | 10.1972 |
| have | 299 | 3726 | 10.096 |
| ignored | 182 | 308 | 10.0737 |
| regulator | 182 | 347 | 10.0636 |
| used | 206 | 1339 | 10.0059 |
| unlawful | 169 | 387 | 9.94632 |
| not | 226 | 2961 | 9.82106 |

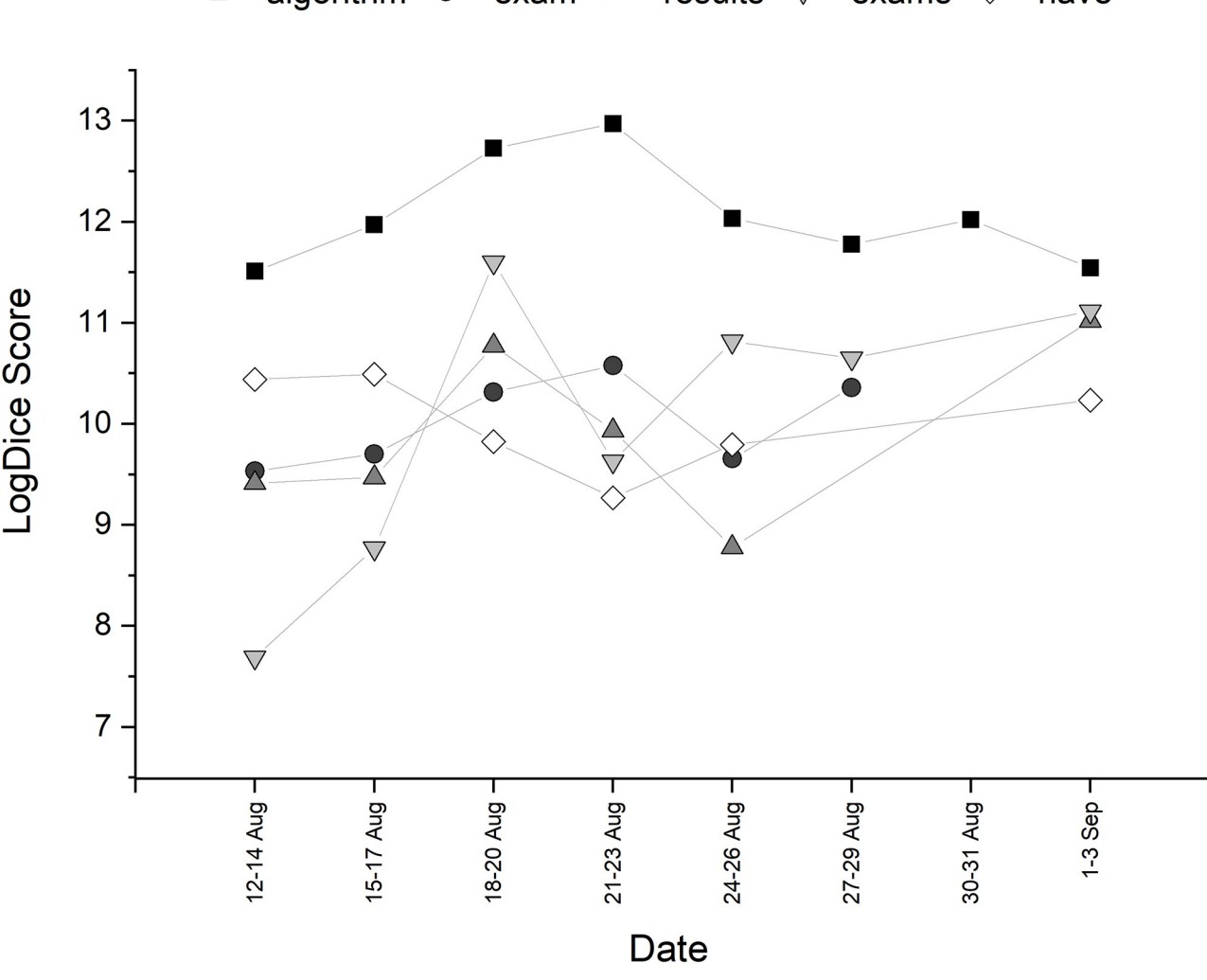

**Fig 4. Temporal trajectory of LogDice scores of collocates of *Ofqual*—Part A.**

August contains 'it's their faith in these one-dimensional metrics that bedevills education', with the possessive pronoun 'their' referring to Ofqual. This hyperbolic use of language to heighten emotion and impact, intensifies the focus on Ofqual as a blameworthy social actor. This is exemplified further in a tweet with 135 engagements on 22nd August, stating 'ofqual [. . .] applied the algorithm'.

In later parts of the corpus, this continues. One tweet with 106 engagements on the 2nd September expresses exasperation with Ofqual by stating 'how did the ofqual people not realise that what they did with the algorithm would not be acceptable'. Ofqual is clearly presented as an implicated social actor here, with the algorithm part of the prepositional subject phrase. This emphasises Ofqual's agency and, thus, implies blame to them. These tweets coincide with Ofqual Chair, Roger Taylor, speaking directly to the Educational Select Committee.

Users also placed agency and blame on Taylor himself through *individualism*. This is seen especially in early September 2020, when Taylor spoke to the Educational Select Committee. As early as 13th August, the day results were released to students, Taylor is actively implicated. In the same tweet that stated that the 'algorithm caused today's chaos', the tweet author goes

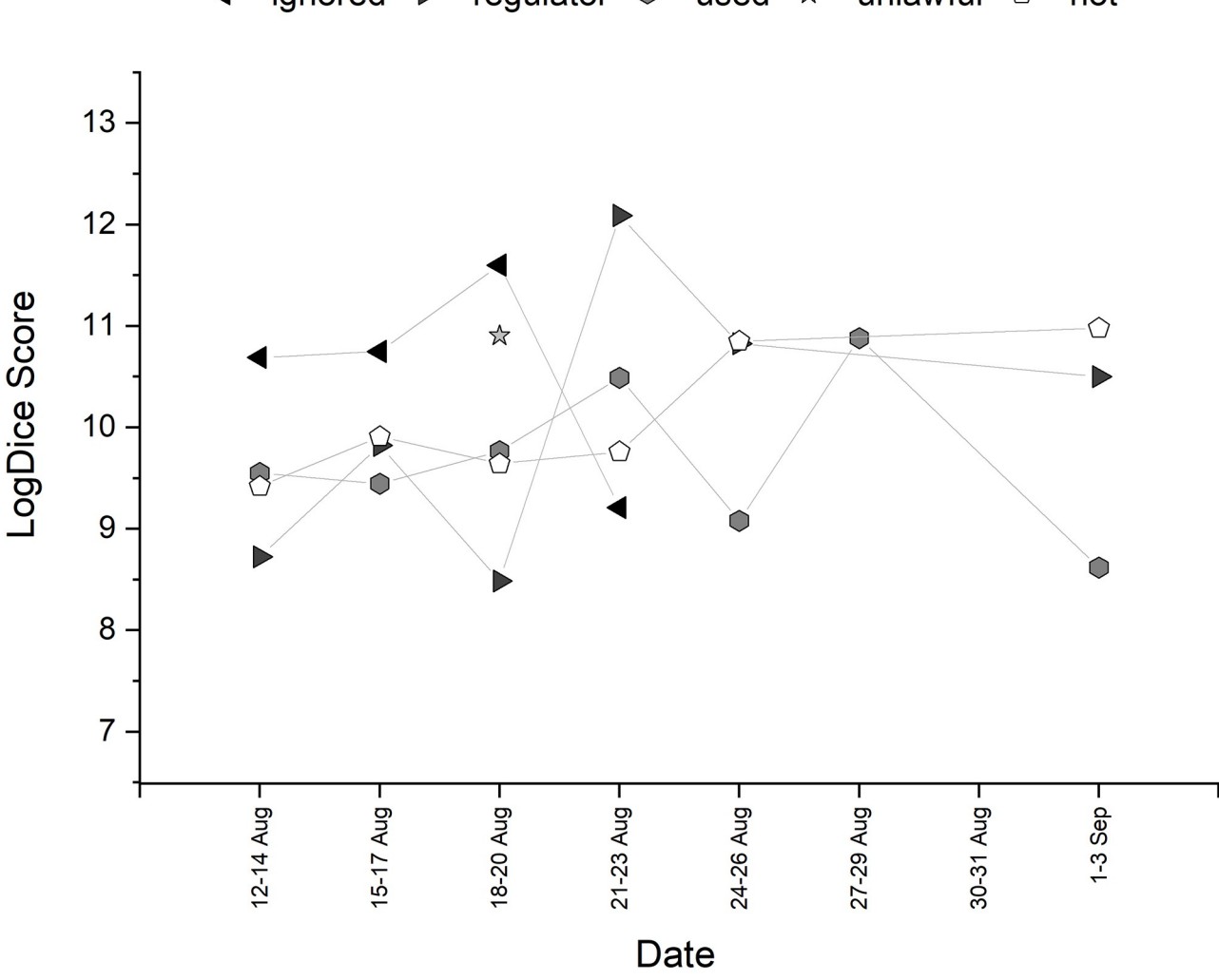

**Fig 5. Temporal trajectory of LogDice scores of collocates of *Ofqual*—Part B.**

on to state that 'ofqual chair roger taylor also chairs the centre for data ethics innovation', which is heavily linked with Dominic Cummings, former advisor to the Boris Johnson. This active construction, and use of the verb 'chairs', which is indicative of status and power, could implicate Taylor, especially with the high engagement with the tweet (3,349 likes and 2,446 retweets). There are other tweets from around a similar time that could place blame on Taylor through agency. For example, one tweet on 16th August states 'roger taylor, [. . .] responsible for the algorithm, flunked his own a levels but was given a "second chance" after passing the entrance exam' (117 engagements). Several verbal phrases in this tweet are attributed to Taylor —including that he is 'responsible' for the algorithm, and, potentially, the failure of the process. Additionally, blame is further implied through the idea that Taylor 'flunked' his exams and 'was given' (a passive construction) a second chance. Similarly to Ofqual, there are times throughout the discourse when the algorithm is attributed to his possession—such as 'benefit from grade inflation under his algorithm' (4164 engagements).

On the 24th August, Taylor is presented in both an active and passive way. For example, a tweet with 518 engagements states 'roger taylor's company was criticised' for failures

concerning algorithms in the past. This passive construction removes the social actor from the construction and foregrounds the importance of Taylor. This is further emphasised by the active role he is given later in the same tweet, when the author writes that 'he's chair of the body charged with overseeing algorithms', and in another tweet that states 'roger taylor chairs both centre for data ethics and innovation (cdei) ofqual'. As well as overtly critiquing Taylor's conflicts of interest by holding multiple senior roles, the use of the lexical item 'chair' (in both noun and verb word classes) reinforces the status, power and responsibility that Taylor has.

On the 2nd September, Taylor appeared at the Educational Select Committee to discuss the algorithm's impact. Tweets placed agency and blame with Taylor. An example includes 'roger taylor [. . .] admits the decision to use an algorithm to award results was a "fundamental mistake" (105 engagements). Taylor is clearly the focal social actor in the construction, with intensity heightened through the use of 'admits'. However, there are other tweets on this date that do implicate Taylor as a blameworthy social actor, but do so by using the word 'tells' in place of 'admits', thus softening the potential blame on Taylor.

To summarise, Ofqual is seen to be presented as a key social actor in this discourse, attracting blame from Twitter users by using active agency and possession. Taylor, here, is seen to be blameworthy through repeated individualism.

### 4.4 The UK government

In this section, the UK government is explored as a potential social actor, focusing on assimilation and individualism for senior government figures. Collocational strength of the top ten words associated with *government* is shown in Table 4. The trajectory of the collocations over time can be seen in Figs 6 and 7. There are words that might be expected to be related to the government (*uk*, *tory*) and also words that are particularly associated with this specific discourse (*ofqual*, *algorithm*, *a-level*). *U-turn*, the word with the highest collocational strength, appears as both a noun ('should the government perform a u-turn'), a verb ('ofqual want the government to u-turn') and, later in the discourse, a noun phrase ('even with the government algorithm u-turn'). The majority attributed the action of the 'u-turn' to the government, as seen in excerpts such as 'the government has u-turned', 'government u turn on exam results' and 'we welcome the government's u-turn'. *After* is frequently used as a prior conjunction to clauses such as these, discussing the need for teacher assessed grades. Unlike the first two entities, this collocation analysis implies the government could be blameworthy.

*Must* is used as a modal verb in a variety of constructions that call on the government to address the situation, such as 'the government must u-turn', 'the government must apply cags'

**Table 4. Collocational strength of *government*.**

| Collocate | Freq | Coll. freq. | logDice |
|---|---|---|---|
| u-turn | 147 | 582 | 11.1546 |
| after | 80 | 764 | 10.1577 |
| must | 56 | 401 | 9.89148 |
| uk | 58 | 548 | 9.83631 |
| ofqual | 176 | 5226 | 9.73726 |
| tory | 45 | 281 | 9.66849 |
| algorithm | 396 | 17225 | 9.43429 |
| a-level | 139 | 6006 | 9.23917 |
| not | 80 | 2961 | 9.18879 |
| have | 93 | 3726 | 9.17913 |

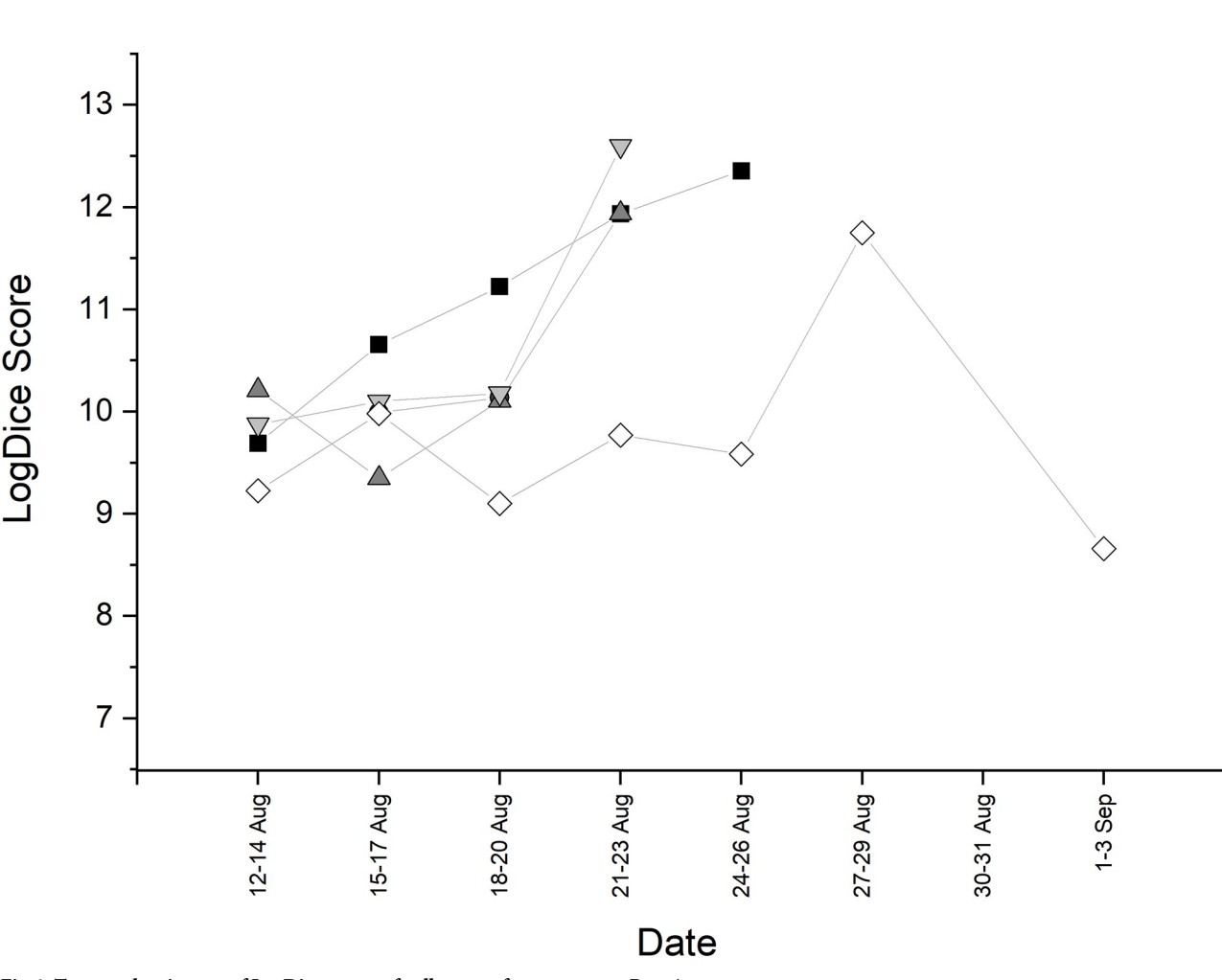

**Fig 6. Temporal trajectory of LogDice scores of collocates of *government*—Part A.**

and 'the government must learn from the shambolic handling of a-level results'. All of these constructions place the government as blameworthy social actors.

Further concordance examination places blame on the UK government as a collective entity, as well as some individual figures. Once again, *assimilation* is found in many constructions. Tweets throughout the discourse refer to the algorithm as 'the government's algorithm', which is expanded upon as a noun phrase by different tweet authors, such as referring to it as the 'hastily-built government algorithm' (665 engagements).

In a direct address to A Level students on 13th August, one author said 'i am sorry this government has failed you' (1599 engagements). Blame is places With the government as the implicated social actor. Further implications of blame could come from the active statements 'government refusing to learn from a level fiasco' (619 engagements) and 'this government really don't like teachers' (1490 engagements). Another tweet stated that the choice of using the algorithm was 'devastating by the uk government' (512 engagements). Although passive, this construction might attribute blame to the government through the foregrounding of the particularly emotive word 'devastating'. This is again seen in 'negatively hurt by the tory

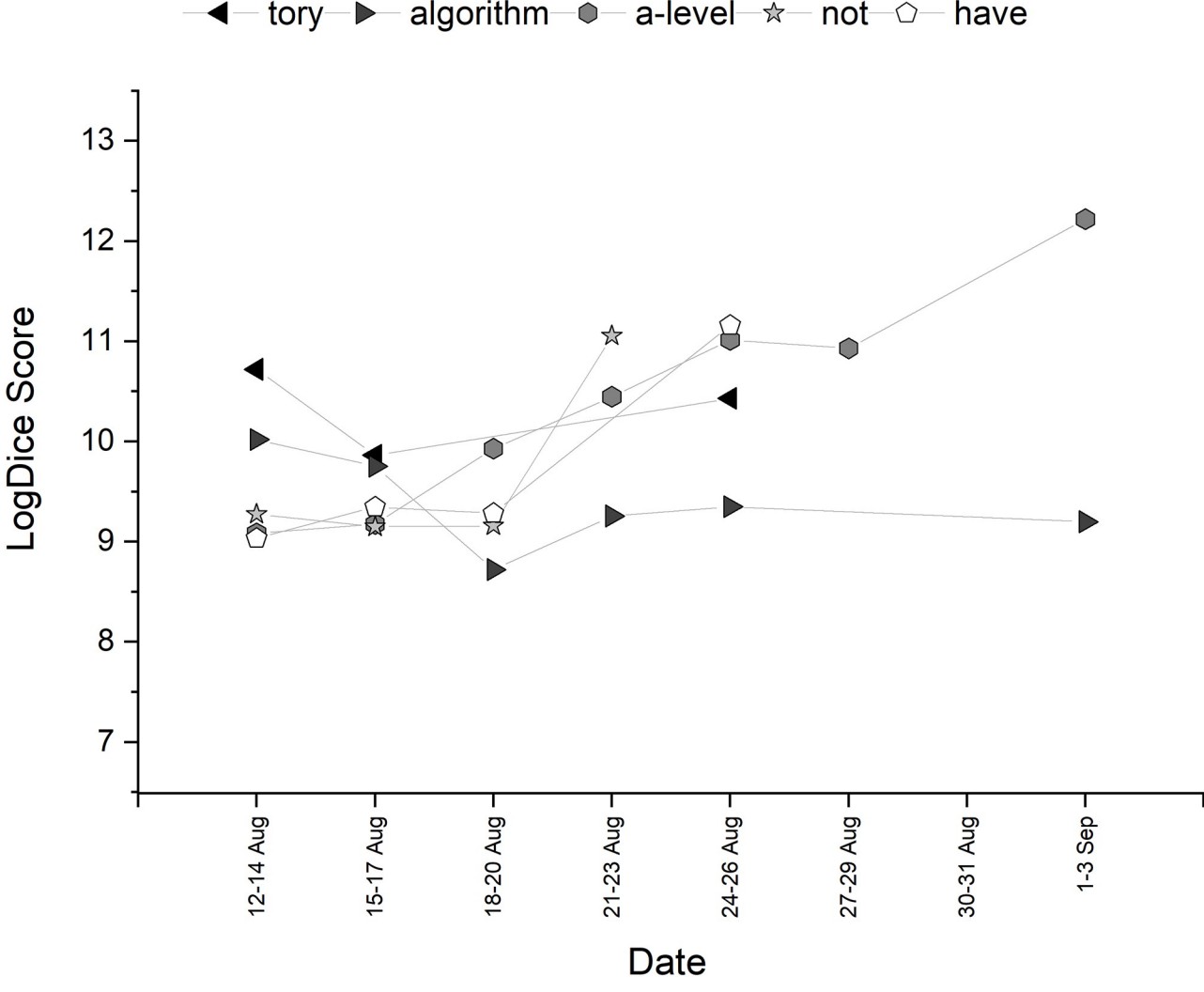

**Fig 7. Temporal trajectory of LogDice scores of collocates of *government*—Part B.**

algorithm' (434 engagements), where emphasis is on the emotion (the 'hurting') rather than government. Although this is *backgrounding*, the implication of blame remains.

There are some instances of support, rather than blame, early on in the discourse, too. A tweet with 362 engagements contains 'the government never trusts teachers but in this v unusual situation it is the fairest way'. The author implies the government is a social actor but in a positive way, despite the verbal phrase 'never trusts' usually being associated with negativity.

Upon the revoking of the use of the algorithm, tweets imply blame is with the government, including one example with 429 engagements that states 'time for the government to hold up their hands'. The implied imperative, the government as the subject of the clause and the colloquialism 'hold up [...] hands' may imply blame. A tweet with 1684 engagements from 18th August says 'the government will blame ofqual', with the active construction perhaps showing that the government is attempting to distract blame from themselves. This is coupled with tweets that expand the possessive noun phrase, such as 'their rigged algorithm' (4105 engagegemts). Later, on 25th August, active constructions further implicate the government, such as

'the government ignored red flags' (35 engagements). This links to the idea that ministers put their 'faith' in the algorithm.

On 26th August 2020, the day that UK Prime Minister Johnson announced that results had been jeopardised by a 'mutant algorithm', Twitter users placed blame with the government. The most engaged-with tweet on this day, which had 5884 likes and 1762 retweets, used a series of rhetorical questions to imply that the government was to blame for the results scandal. Part of the tweet reads, 'who set the parameters for ofqual's algorithm? ministers! who didn't ask the right questions? ministers! who didn't ask for a simulation of the impact? ministers!! so who should resign?' This tweet's use of effective tripling as a rhetorical device is noteworthy, but it also has aspects of agency to explore. The interrogative pronoun 'who' could be substituted for the government (or 'ministers' in this case), making them an implied active social actor in the fault of the algorithm. Although the responses to the tweet were not part of the original dataset, there were other tweets within the dataset that linked the same BBC article, thus acting as a springboard for conversation and framed contextually around this specific piece of information. These tweets presented the government as implicated social actors.

Once again, there are individual social actors within this body, explored as *individualism*. Firstly, there are specific instances where blame is attributed to UK Prime Minster Boris Johnson. Upon the release of results, structures in tweets indicated that he had ownership of the algorithm, such as 'clever boris' algorithm' (96471 engagements), implying blame is with Johnson. Additional tweets also indicate blame with Johnson, specifically on 26th August. One user tweeted about Johnson that 'he can't wriggle out of responsibility with bluster and distortion' (32 engagements). This presents Johnson as the active social actor and the verb phrase 'wriggle out' may indicate he is to blame.

There are also a number of tweets that discuss Gavin Williamson, UK Education Secretary of State at the time of the A Level results in 2020. On the day of the government u-turn, one tweet stated that Williamson has 'signed off on' the algorithm (700 engagements), showcasing him as an blameworthy social actor and decision-maker. On 18th August, after the reversal, constructions included 'williamson is trying to blame ofqual' and 'he admits he didn't even bother checking it' (224 engagements). These constructions show his active agency. However, Williamson is also presented in passive constructions, with one tweet with 524 engagements saying that he 'was badly advised'. This reduces blame towards Williamson, especially through the obscuring of an unknown social actor in the construction through *exclusion*.

In summary, the findings here indicate that elements of blame through active agency and social action for the government can be derived from the tweets. Passive constructions use emotive language that still imply blame is with the government). There are times when assimilation occurs and, as the discourse continues, individualism is more apparent for Johnson and Williamson.

## 4.5 Students

Collocational strength of the top ten words associated with *students* is shown in Table 5. Again, there are anticipated semantically-related words present (*a-level*, *grades*, *gcse*, *england*). Many of the occurrences of *their* relate to how well teachers know their students (seemingly in retaliation to the decision to use an algorithm to calculate grades, rather than teachers, and discussions about their futures in the wake of the decisions made.

The strength of the relationship of *students* and *downgraded* can also be examined. These are a mix of passive ('students getting downgraded results by some algorithm') and active ('algorithm that downgraded many disadvantaged students') constructions, where students were the object in either. There were instances where the verb 'downgraded' was intransitive

**Table 5. Collocational strength of *students*.**

| Collocate | Freq | Coll. freq. | logDice |
|---|---:|---:|---:|
| a-level | 651 | 6006 | 11.23 |
| their | 395 | 2757 | 11.1663 |
| grades | 373 | 3475 | 10.9105 |
| gcse | 238 | 1344 | 10.8521 |
| have | 336 | 3726 | 10.7039 |
| downgraded | 155 | 914 | 10.3885 |
| england | 132 | 767 | 10.2139 |
| many | 115 | 613 | 10.0773 |
| all | 139 | 1425 | 10.0488 |
| given | 108 | 473 | 10.0458 |

and the social actor performing the action was not included in the tweet ('40% of a-level students being downgraded'). While this reduces potential blame for students, it does not implicate another social actor. It is also important to note that this is another example of *assimilation*. Upon further CDA examination, it appeared that students were presented as passive in the majority of constructions, regardless of the verb used, including *given* when the decision was reversed ('students in england will be given grades estimated by their teachers'— a tweet with many retweets). This may suggest that students are not as heavily implicated.

## 5 Discussion

In the following, we discuss the implications of blame being attributed to the algorithm itself, Ofqual and the UK government through the combination of collocation, transitivity and social action analysis. Although these are three different aspects, in this study they are explored in an intertwined way. We relate this to previous research into the algorithm and the A Level results of 2020 to contribute to existing analysis concerning blame and responsibility for the issuing of results. After, we consider how the results work in a complementary way to NLP-based computational linguistic findings, building on our previously identified research gap.

### 5.1 Blame for the A Level results

Through the analysis of transitivity in concordance lines, collocation and CDA, underpinned by SAR, it was possible to see how blame is attributed to social actors throughout this Twitter discourse. The algorithm itself is most commonly presented as having active agency. The tweets seen that support this seem to imply that the algorithm is a social actor, despite its inanimate state, and so blame is shifted to the algorithm. Tweets imply that the algorithm is able to make decisions independently. This is in line with expectations of agency and blame that are outlined by Richardson et al. [60] and personalisation by Van Leeuwen [11].

Through personification and agency metaphor, the algorithm is depicted as carrying out human-like actions. This appears to support the idea of Goatly [63] that this is done for increased dramatic effect and implies the algorithm has the capacity to make independent decisions, such as removing pathways to university.

Although less frequently, there are also times where the algorithm is included in passive constructions. This is especially true when the algorithm is being referred to as being used by an unknown social actor, thus shielding the 'user', and may take agency away from the algorithm and obscure blame. There are times when more intense verbs are used in passive constructions, still implicating the algorithm. This relates to the notions of agency specified by

Clark [61] and could seen to be obscuring agency through *backgrounding*, according to principles of SAR [11].

However, considering verb choices, there are passive constructions that contain the verbs 'assigned' and 'graded'. Thus, a small portion of tweets using passive constructions appear to imply that the algorithm can still be blamed. This can be categorised as agency metaphor according to Morris et al. [62].

This builds upon existing research that Bhopal and Myers found that students thought that the algorithm's result generation was unfair, thus implicating the algorithm [26] and ties into the potential backlash against algorithms that was reported to have occurred—and predicted to intensify—by Kolkman [27] and Hecht [28]. This, in turn, supports one of our other findings: that students were not blamed through agency and transitivity in this Twitter discourse due to their passive presentation.

The UK government and the regulation body Ofqual were also presented as responsible social actors by Twitter users. For both social actors, active statements were seen that could implicate them as agents of blame. This was less frequent than the algorithm was implicated at the start of the sampled discourse and more frequent towards the end of the discourse. *Assimilation* and *individualism* were both seen here.

Some tweets show how blame is attributed to social actors through the possession of another. For example, Ofqual and the UK government were, in many tweets, seen to be the owners of the algorithm, which implicates that they are to blame for the failures of the algorithm. This occurs throughout the discourse, especially on dates of significant events, such as the algorithm belonging to Roger Taylor on the date he appeared at the Educational Select Committee, the algorithm belonging to Boris Johnson on the date he called it a 'mutant algorithm', and the algorithm belonging to Gavin Williamson on the date of the u-turn. The idea of another entity possessing the implicated entity of the algorithm also blurs blame. The examination of how context affects language plays a crucial role in finding how blame is expressed through transitivity and, also, possession [53].

## 5.2 Use of corpus linguistics and critical discourse analysis to complement NLP-based computational methods like sentiment analysis

One of the aims of this study was to see how the qualitative findings CL and CDA, in addition to statistical collocation measures, provided further nuance to the quantitative findings from using sentiment analysis.

Overall, using the sentiment trajectory from the study by Heaton et al. [5] provided a sound starting point for analysis. An example of this is the analysis conducted on 26th August, where the examination of VADER sentiment analysis pinpoints 26th August as the date with the largest sentiment change and the lowest sentiment value in the discourse. Through using CL and CDA, it was clear that the majority of blame—through active agency, agency metaphors, hyperbole, possession, assimilation and individualism—on this date was directed towards the UK government and Boris Johnson. This was the date he declared the algorithm to be 'mutant'. The combination of analyses through may suggest that Johnson's actions implicated him as responsible for the failure of the algorithm's deployment due to the fact that the previous sentiment scores were low and tweet authors portrayed him as an implicated social actor.

CL was used primarily to identify potential social actors of blame and uncover patterns of transitivity [39]. Combining these analytical perspectives enhances the findings beyond sentiment analysis.

There were, however, some issues with the data collection process. Upon reviewing tweets, it was clear that there were many replies to tweets that form part of the discourse. But, due to

the specific parameters of the search criteria used to collect this data, these replies were not part of the dataset. This potentially limits findings, especially as CDA is underpinned by the analysis of interaction between others [53]. However, other tweets used the same news articles to provide context to their tweets. This is still a response to a main source and connects tweets to one another, therefore mitigating some of these shortcomings.

Above all, this demonstrates that the combination of CL and CDA continues to be a suitable mechanism to be deployed on Twitter discourses surrounding social and topical issues [13–15]. It also demonstrates value for a combination of qualitative and quantitative measures being used to analyse social media [65]. This echoes the findings of previous studies that have done this successfully with different qualitative methods [35, 36] and showcases that this combination can be applied to Twitter discourses too. Ultimately, using CL and CDA provided a better lens to explore urgent social ideas and, in our case, blame and social actors [55].

## 5.3 Limitations and future work

There are some things that limit the success of the study. As there were over 18,000 tweets, it is not possible to have examined all of these in great detail [67]. Although the use of CL may have mitigated this somewhat, even more insight may be waiting to be unearthed in this dataset. As previously expressed, the search criteria used to form the initial dataset may be missing important aspects of the discourse due to its strict lexical conditions. Finally, using CDA means that we approached the analysis with our own biases and subjective perspectives, potentially questioning the validity of the insights [64, 68].

When considering future work, there is potential to use CL and CDA to investigate related threads or themes. For example, we could enhance this exploration by investigating thematisation (which would link to the latent topics found using computational linguistics) [82] and the use of structural-functional linguistics and social-semiotics. This allows greater depth of research into the views expressed about the algorithm and could be done by multiple researchers to mitigate subjective biases. On a related note, a further suggestion may be to continue to use SAR to examine how the different social actors interact with one another.

Another suggestion is to improve the approach of 'quantitative first, qualitative second' into a more iterative cycle. Considering principles of iterative data science, such as the 'epicycles of data analysis' [83], a process could focus on the cyclical development of expectations, analysis of data, and matching of expectations to data, which repeats. This might mitigate not being able to analyse the replies excluded from the original dataset. In this model, the discourse becomes a 'moving feast', where NLP-based tools can then be re-deployed to capture replies to key tweets, which are further analysed using CL and CDA. Similarly, Social Network Analysis could be used with NLP and CL approaches to explore language patterns in this discourse, in a similar way to McGlashan and Hardaker [84].

## 6 Conclusion

The sociolinguistic findings reported and discussed in this contribution show that, through using CL and CDA, many Twitter users blamed the algorithm as a standalone social actor for the A Level results. This reaction was expressed through active agency, including agency metaphor (such as 'that algorithm is going to screw you') and personalisation of the algorithm (such as 'the job of the algorithm').

Additionally, the UK government and Ofqual, and devolved social actors within these organisations like Taylor and Johnson, were also blamed by Twitter users through similar constructions and elements of possession (such as 'benefit from grade inflation under his algorithm'). This was seen less frequently at the start of the discourse and more frequently

towards the end. This was mainly done through assimilation in earlier tweets and individualism in later tweets.

Furthemore, passive constructions could be seen for all of these social actors, with some indicating more blame than others (such as 'the algorithm used by ofqual'). Techniques to obscure and shift blame were also seen, like backgrounding (such as 'devastating by the uk government') and exclusion (such as 'he was badly advised').

Ultimately, although it could not be determined which social actor out of the algorithm, Ofqual and the government was blamed the most, we conclude that these entities were presented as blameworthy social actors throughout the discourse. As well as providing insights into the online response to this particular event, there is potential for broader impact too. Despite the disruption of the pandemic coming to an end in the UK, this contribution provides insights into how members of the public may react to future decision-making algorithm interventions.

In addition, the methodological conclusions illustrate how CL and CDA can be used in a complementary way to NLP-based computational linguistic tools like sentiment analysis. More specifically, using quantitative data as starting points allows for more focused qualitative analysis. For example, the previously reported significant negative shifts in sentiment coincided with more authors suggesting blame was with the UK government and Boris Johnson. To ensure the application of 'epicycles of data science' creates an iterative computational and discursive methodological process, a more in-depth investigation of blame attribution and expression is needed.

## Supporting information

**S1 Dataset. Dataset used in the study.** Including search term, tweet ID, timestamp, number of favourites and number of retweets.
(CSV)

## Author Contributions

**Conceptualization:** Dan Heaton, Elena Nichele, Joel E. Fischer.

**Data curation:** Dan Heaton.

**Formal analysis:** Dan Heaton, Elena Nichele.

**Funding acquisition:** Dan Heaton.

**Investigation:** Dan Heaton.

**Methodology:** Dan Heaton, Elena Nichele.

**Project administration:** Dan Heaton.

**Resources:** Dan Heaton.

**Software:** Dan Heaton.

**Supervision:** Elena Nichele, Jeremie Clos, Joel E. Fischer.

**Validation:** Dan Heaton.

**Visualization:** Dan Heaton.

**Writing – original draft:** Dan Heaton, Elena Nichele.

**Writing – review & editing:** Dan Heaton, Elena Nichele, Jeremie Clos, Joel E. Fischer.

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
