## [Decision Letter · Decision Letter 0]

9 May 2023

PONE-D-23-08581"The Algorithm Will Screw You'" Blame, Social Actors and the 2020 A Level Results Algorithm on TwitterPLOS ONE

Dear Dr. Heaton,

Thank you for submitting your manuscript to PLOS ONE. After careful consideration, we feel that it has merit but does not fully meet PLOS ONE’s publication criteria as it currently stands. Therefore, we invite you to submit a revised version of the manuscript that addresses the points raised during the review process.

We look forward to receiving your revised manuscript.

Kind regards,

Michal Ptaszynski, PhD

Academic Editor

PLOS ONE

Journal Requirements:

Reviewers' comments:

Reviewer's Responses to Questions

**Comments to the Author**

1. Is the manuscript technically sound, and do the data support the conclusions?

Reviewer #1: Yes

Reviewer #2: Yes

2. Has the statistical analysis been performed appropriately and rigorously? 

Reviewer #1: Yes

Reviewer #2: Yes

3. Have the authors made all data underlying the findings in their manuscript fully available?

Reviewer #1: Yes

Reviewer #2: Yes

4. Is the manuscript presented in an intelligible fashion and written in standard English?

Reviewer #1: Yes

Reviewer #2: Yes

5. Review Comments to the Author

Reviewer #1: Thanks for presenting an interesting study.

The dataset of tweets is almost three-year-old. What is the significance of analyzing this old dataset? Do your conclusions still hold in 2023?

A better explanation and subsequent implications are needed for the drop in the VADER line around August 2020 in Figure 1.

I appreciate the use of discourse analysis.

Please explain what you mean by the agency. Do social actors always act as a social agency? A literature review on agency and situating your study against past research will be useful.

How do different social actors interact with each other, and what are the implications of their interaction for your study?

Are you sure about the text in the Acknowledgments section?

Reviewer #2: This is an interesting manuscript that touches upon a very important topic. In the following, I would like so hare some thoughts the authors might consider to possibly further improve the quality of the paper.

Introduction

The authors state “there is a research gap regarding public views expressed on Twitter […]”. I agree, but why does this need to be addressed? The following text does not explain this.

The rest of the introduction is very much about the chosen methodological approach. This is understandable. However, to underline the practical relevance of the authors’ approach, I would suggest spending some more time on contextualizing the methods in the wider, practical discourse. I understand that this is what the next section does. I think it would be an alternative option to already mention more about this earlier.

Context of the 2020 A Level Algorithm

I generally like this section. However, towards the end I think the authors should consider referring back to why and how their methods are going to add more information about the discourse. Moreover, the authors state “yet limited research into how social media users reacted to 101 the scandal, thus providing motivation for our research”. Again, I agree. But for the purpose of the paper, I think the authors should spend more time in explaining why this is important and what additional insights it can provide. The next section does this – I just think that the transitions could be improved and made a bit more fluent.

Related Work

“Using CL and CDA, underpinned by SAR, it will be possible to ultimately

contribute to filling the gap previously identified in the literature.“ Why and how?

NLP-Based Computational Linguistics to Examine Social Media

It seems that this section already provides results from the collected data. However, this is really not clear to me. Overall, this section is difficult to read and follow.

Using Corpus Linguistics to Examine Social Media

I like this section.

Using Critical Discourse Analysis to Examine Social Media

I like this section and how the authors describe how this approach can adhere to the gaps of the other approach. Yet, the link to social media and Twitter is rather short and I would like to suggest that the authors spend more time on expanding the paragraph on page 7, lines 277-287.

Research Gap

I think Social Actor Representation (SAR) and Social Action Theory (SAT) need to be mentioned earlier in the manuscript. This seems to be crucial for the paper. But only surface just before the Methods section. Right now, the link to the method and why it is important are too short and sometimes just constructed with one sentence.

Method

Good. I just think that Table 1 is not positioned well in this section.

Results

“Based on this first list, four potential social 379 actors (the algorithm, Ofqual, the government and students) were investigated through 380 the examination of collocational strength and CDA.” I am struggling a bit to consider “the algorithm” as a social actor. I think I know what the authors are referring to, but maybe they could spend some more time making an argument that this link can be made. The discussion touches upon this. But, in my opinion, it might be a bit late then.

The Algorithm

This is an interesting section. However, I think it would be really beneficial to add another Figure that shows the frequency of the collocations across time. I think it would add another very valuable layer to the description in the text. Similarly, while the next two sections are equally interesting, I wonder whether the authors could consider a more visual representation of their findings as they have a clear timeline going through the analyses.

Discussion

Good

Limitations and Future Work

I was a bit surprised not to see any reference to Social Network Analyses, which particularly in combination with CL is becoming more common in research. Additionally, while I understand the authors’ criticism of sentiment analyses, I think that it would have been interesting to more carefully combine this with their CL approach of collocations and POS.

Conclusion

The methodological conclusion is understandable. The overall conclusion of “blaming a social actor” is of course more difficult, but also less clear in the description.

6. PLOS authors have the option to publish the peer review history of their article (what does this mean?). If published, this will include your full peer review and any attached files.

Reviewer #1: No

Reviewer #2: No

---

## [Author Response · Author response to Decision Letter 0]

20 May 2023

Comment Page(s) Action

The dataset of tweets is almost three-year-old. What is the significance of analyzing this old dataset? Do your conclusions still hold in 2023? 20 We thank the reviewer for this comment. Whilst the disruption of the pandemic is coming to an end, this paper gives insight into how members of the public may react to future decision-making algorithm interventions. We have included this as a statement in the conclusion.

A better explanation and subsequent implications are needed for the drop in the VADER line around August 2020 in Figure 1. 4 More detail has been added here – this is most likely caused by ‘mutant’ holding negative sentiment score, combined with an increase in the number of negative responses.

Please explain what you mean by the agency. Do social actors always act as a social agency? A literature review on agency and situating your study against past research will be useful. 7-8 Thank you for this suggestion. We define agency on page 7 of the manuscript. We have expanded on the referenced literature on page 9 relating to using agency alongside CL and CDA to explore social media discourses.

How do different social actors interact with each other, and what are the implications of their interaction for your study? 20 We have considered this carefully and have concluded that the aim of the paper is to look at how the social actors are represented, rather than how they interact with one another. We have, instead, offered a future work suggestion for exploring this.

The authors state “there is a research gap regarding public views expressed on Twitter […]”. I agree, but why does this need to be addressed? The following text does not explain this. 1-2 Further explanation and detail has been added after this sentence to explain how using social media data can add to the bigger picture of the public’s response to the algorithm.

The rest of the introduction is very much about the chosen methodological approach. This is understandable. However, to underline the practical relevance of the authors’ approach, I would suggest spending some more time on contextualizing the methods in the wider, practical discourse. I understand that this is what the next section does. I think it would be an alternative option to already mention more about this earlier. 2-3 We have added a sentence to make reference to contextualising the methods in the wider, practical discourse and alluded to how these will be explored later in the paper.

However, towards the end of the ‘Context of the Algorithm’ section, I think the authors should consider referring back to why and how their methods are going to add more information about the discourse. 3 We agree and have added a short paragraph explaining this.

Moreover, the authors state “yet limited research into how social media users reacted to 101 the scandal, thus providing motivation for our research”. Again, I agree. But for the purpose of the paper, I think the authors should spend more time in explaining why this is important and what additional insights it can provide. The next section does this – I just think that the transitions could be improved and made a bit more fluent. 3 This has been addressed in the additional paragraph also.

“Using CL and CDA, underpinned by SAR, it will be possible to ultimately

contribute to filling the gap previously identified in the literature.“ Why and how? 4 This sentence has been extended to add further clarity.

NLP-Based Computational Linguistics to Examine Social Media: It seems that this section already provides results from the collected data. However, this is really not clear to me. Overall, this section is difficult to read and follow. 4-5 More discourse markers have been added to ensure clarity in this section.

Using Critical Discourse Analysis to Examine Social Media: I like this section and how the authors describe how this approach can adhere to the gaps of the other approach. Yet, the link to social media and Twitter is rather short and I would like to suggest that the authors spend more time on expanding the paragraph on page 7, lines 277-287. 7 Thank you for this comment. We have expanded this section with more specific findings from these case studies to illustrate the depth of understanding that CDA can uncover.

Research Gap: I think Social Actor Representation (SAR) and Social Action Theory (SAT) need to be mentioned earlier in the manuscript. This seems to be crucial for the paper. But only surface just before the Methods section. Right now, the link to the method and why it is important are too short and sometimes just constructed with one sentence. 8 We thank the reviewer for this observation and suggestion. Upon reflection, we have kept the main part of SAR and SAT in this section. However, we have introduced and provides information about these concepts earlier in order to foreground their importance in our work. 

Method: Good. I just think that Table 1 is not positioned well in this section. 10 Agreed. This has been adjusted so it is in section 4.

“Based on this first list, four potential social 379 actors (the algorithm, Ofqual, the government and students) were investigated through 380 the examination of collocational strength and CDA.” I am struggling a bit to consider “the algorithm” as a social actor. I think I know what the authors are referring to, but maybe they could spend some more time making an argument that this link can be made. The discussion touches upon this. But, in my opinion, it might be a bit late then. 10 Additional rationale has been included here: as all of these words are nouns that can be presented actively in a grammatical construction, they are all capable of being a social actor. 

The Algorithm: This is an interesting section. However, I think it would be really beneficial to add another Figure that shows the frequency of the collocations across time. I think it would add another very valuable layer to the description in the text. Similarly, while the next two sections are equally interesting, I wonder whether the authors could consider a more visual representation of their findings as they have a clear timeline going through the analyses. Figures Linked scatter diagrams – grouped on three day intervals – have been inserted to show the trajectories of the LogDice scores over time. These have been split into part A and part B to prevent cluttered figures.

Limitations and Future Work: I was a bit surprised not to see any reference to Social Network Analyses, which particularly in combination with CL is becoming more common in research. Additionally, while I understand the authors’ criticism of sentiment analyses, I think that it would have been interesting to more carefully combine this with their CL approach of collocations and POS. 18-19 We have considered this and feel that Social Network Analyses would be an example of future work. Thus, we have included reference to this in this section.

Conclusion: The methodological conclusion is understandable. The overall conclusion of “blaming a social actor” is of course more difficult, but also less clear in the description. 20 Discourse markers have been added to make the conclusion clearer: all three of the social actors explored are blameworthy, although it is not possible to confirm which one was blamed the most.

---

## [Decision Letter · Decision Letter 1]

2 Jul 2023

"The Algorithm Will Screw You'" Blame, Social Actors and the 2020 A Level Results Algorithm on Twitter

PONE-D-23-08581R1

Dear Dr. Heaton,

We’re pleased to inform you that your manuscript has been judged scientifically suitable for publication and will be formally accepted for publication once it meets all outstanding technical requirements.

Kind regards,

Michal Ptaszynski, PhD

Academic Editor

PLOS ONE

Additional Editor Comments (optional):

Reviewers' comments:

Reviewer's Responses to Questions

**Comments to the Author**

1. If the authors have adequately addressed your comments raised in a previous round of review and you feel that this manuscript is now acceptable for publication, you may indicate that here to bypass the “Comments to the Author” section, enter your conflict of interest statement in the “Confidential to Editor” section, and submit your "Accept" recommendation.

Reviewer #1: All comments have been addressed

Reviewer #2: All comments have been addressed

2. Is the manuscript technically sound, and do the data support the conclusions?

Reviewer #1: Yes

Reviewer #2: Yes

3. Has the statistical analysis been performed appropriately and rigorously? 

Reviewer #1: Yes

Reviewer #2: Yes

4. Have the authors made all data underlying the findings in their manuscript fully available?

Reviewer #1: Yes

Reviewer #2: Yes

5. Is the manuscript presented in an intelligible fashion and written in standard English?

Reviewer #1: Yes

Reviewer #2: Yes

6. Review Comments to the Author

Reviewer #1: Thanks for incorporating reviewer feedback. I am satisfied with the revised version of your manuscript.

Reviewer #2: I would like to thank the authors for carefully considering the feedback and making applicable adjustments where suggested. There remain some small issues (e.g. seeminglz missing references - line 196). Other than that, this is a nice piece of research.

7. PLOS authors have the option to publish the peer review history of their article (what does this mean?). If published, this will include your full peer review and any attached files.

Reviewer #1: No

Reviewer #2: No

---

## [Editor Report · Acceptance letter]

5 Jul 2023

PONE-D-23-08581R1 

“The Algorithm Will Screw You”: Blame, Social Actors and the 2020 A Level Results Algorithm on Twitter 

Dear Dr. Heaton:

I'm pleased to inform you that your manuscript has been deemed suitable for publication in PLOS ONE. Congratulations! Your manuscript is now with our production department. 

Kind regards, 

on behalf of

Dr. Michal Ptaszynski 

Academic Editor

PLOS ONE